# Rapid Assessment of Microbial Quality in Edible Seaweeds Using Sensor Techniques Based on Spectroscopy, Imaging Analysis and Sensors Mimicking Human Senses

**DOI:** 10.3390/s22187018

**Published:** 2022-09-16

**Authors:** Anastasia E. Lytou, Panagiotis Tsakanikas, Dimitra Lymperi, George-John E. Nychas

**Affiliations:** Laboratory of Microbiology and Biotechnology of Foods, Department of Food Science and Human Nutrition, School of Food and Nutritional Sciences, Agricultural University of Athens, 118 55 Athens, Greece

**Keywords:** marine algae, spoilage, FT-IR, multispectral imaging, e-nose, machine learning

## Abstract

The expansion of the seaweed aquaculture sector along with the rapid deterioration of these products escalates the importance of implementing rapid, real-time techniques for their quality assessment. Seaweed samples originating from Scotland and Ireland were stored under various temperature conditions for specific time intervals. Microbiological analysis was performed throughout storage to assess the total viable counts (TVC), while in parallel FT-IR spectroscopy, multispectral imaging (MSI) and electronic nose (e-nose) analyses were conducted. Machine learning models (partial least square regression (PLS-R)) were developed to assess any correlations between sensor and microbiological data. Microbial counts ranged from 1.8 to 9.5 log CFU/g, while the microbial growth rate was affected by origin, harvest year and storage temperature. The models developed using FT-IR data indicated a good prediction performance on the external test dataset. The model developed by combining data from both origins resulted in satisfactory prediction performance, exhibiting enhanced robustness from being origin unaware towards microbiological population prediction. The results of the model developed with the MSI data indicated a relatively good prediction performance on the external test dataset in spite of the high RMSE values, whereas while using e-nose data from both MI and SAMS, a poor prediction performance of the model was reported.

## 1. Introduction

In the last two decades (2005 onwards) annual global seaweed production increased from 13.5 to 30.4 million tons per year, making it one of the fastest growing food sectors with a projected global seaweed production in 2050 of 500 million tons [1,2]. So far, various studies have focused on the utilization of seaweed-derived polysaccharides for the development of packaging coatings and films because of their renewability and sustainability. Alginates, agar and carrageenan are seaweed-derived polysaccharides that are widely used in the development of coatings and films due to their gelling ability.

It is only recently that seaweed farming is gaining increased attention for the promotion and monitoring of climate and environmentally friendly bio-economy development. Indeed, there have been research studies aimed at achieving new ecological and efficient vegetable culture methods, such as hydroponics or seaweed farming [3,4]. In comparison with hydroponic cultures, which cause a negative environmental impact by putting pressure on freshwater reserves, seaweed cultivation only requires the seawater available in the open ocean. Although seaweed is an under-exploited crop, it has recently been considered a sea vegetable. The majority of seaweed is not only edible but also a great source of macro- and micro-nutrients. Although these products are not very popular, there are areas around the world, i.e., Japan, where one fifth of daily meals have seaweed or seaweed compounds [5]. It is only recently that seaweed has been inserted into Western diets, and they are gaining more interest due to the development of the health food industry and the transition into nutraceuticals [6,7].

The available information regarding the parameters dealing with the quality and safety of these products is limited. Furthermore, there are significant gaps in the regulations related to seaweed quality and safety from a microbiological perspective, while a limited number of studies are dealing with the determination/assessment of the quality and/or safety of these products. In general, the shelf-life of fresh seaweed ranges from 3 to 14 days [8,9] depending on several parameters such as environmental conditions in the period around harvest, treatments during and after harvest, early or late harvest, etc. Thus, apart from stabilizing the product to avoid food losses and/or food waste, i.e., spoilage, it is also necessary to develop rapid analytical technologies for the estimation of the microbiological quality and freshness. These approaches enhance sustainability and contribute to the marketing of higher quality products by increasing—at the same time—the profitability in the aquaculture industry. This issue is not related exclusively to the seaweed industry, but it broadly concerns the food sector in general. The last few decades there has been a huge effort from stakeholders to investigate rapid methods that are suitable for online, real-time food quality/safety assessments [10,11]. Towards this direction, several research groups have investigated the potential of different analytical technologies to estimate various parameters related to food quality. The application of e-nose and e-tongue systems in food quality assessment, with emphasis on commonly used pattern recognition algorithms, have been thoroughly discussed [12], while the use of FT-IR spectroscopy and multispectral imaging in the evaluation of different products of animal and plant origin has also been investigated [13,14,15].

Given that a large amount of data can be generated by applying the aforementioned technologies, data analysis is of critical importance for the utilization of the complete information derived from these techniques. Data obtained from these sensors in combination with machine learning algorithms (e.g., partial least square regression—PLS-R, support vector machines—SVM, tree-based algorithms—random forests, extra trees, etc.) can successfully contribute to the prediction of several variables relevant to food quality as well as the discrimination of different food products based on their differences or similarities [16].

This study aimed to investigate the potential of FT-IR spectroscopy, multispectral imaging and e-nose analysis combined with machine learning models to estimate microbial counts in edible brown marine algae, *A. esculenta*.

## 2. Materials and Methods

### 2.1. Sample Collection

Seaweed samples from two aquaculture sites located in different countries (United Kingdom—Scotland and Ireland) were analyzed. *A. esculenta* from Scotland was cultivated at the Port-a-Bhuiltin seaweed farm operated by the Scottish Association for Marine Science (SAMS, Dunbeg, Scotland), while seaweed from Ireland was cultivated at a pilot scale site, Lehanagh Pool, operated by the Marine Institute (MI) on the west coast of Ireland. Thalli of appropriate size were selected, cut and placed into clean containers. An amount of ca., 4 kg, was transferred into plastic bags and stored at −20 °C until shipping. On the day of shipping, the plastic bags were placed into polystyrene boxes and shipped to the Laboratory of Food Microbiology and Biotechnology of the Agricultural University of Athens within 48 h, while the samples remained frozen during the transfer. Two different batches of seaweed originating from two (2019 and 2020) and three (2019, 2020 and 2021) different years in the case of SAMS and MI, respectively, were examined. Each batch of seaweed corresponded to a specific harvest. The optimum harvest period was decided by the aquaculture site operators based on the appropriate size and overall appearance of the products.

The frozen samples were allowed to thaw overnight at 0–1 °C and subsequently divided aseptically into 50 g portions, placed in polystyrene trays and stored aerobically at different temperature conditions. Products from SAMS harvested in 2019 were stored at 5 and 15 °C, while those from 2020 were stored at 0, 5, 10 and 15 °C. MI’s samples were stored at 5 and 10 °C (2019); 0, 5, 10 and 15 °C (2020); and 5 °C (2021). At this point, it should be noted that the samples were used as provided, without applying any other processing (e.g., drying) except for thawing. Four (4) samples of each seaweed species and storage temperature were analyzed microbiologically at specific time intervals. In order to avoid any changes in products caused by re-freezing and re-thawing seaweeds, the experiment for each year and pilot site were conducted one time.

### 2.2. Microbiological Analysis

For the estimation of total aerobic counts (TAC), fifteen grams of seaweed were transferred aseptically to a stomacher bag, diluted ten times using sterile maximum recovery diluent (MRD) and homogenized in a stomacher (Lab Blender, Seward Medical, London, UK) for 120 s at room temperature. The homogenate was then serially diluted in testing tubes and 0.1 mL of the appropriate dilution was spread in duplicate on marine agar (10,059.00, Condalab, Spain). After incubation at 30 °C for 48 h, colonies were enumerated and their counts were logarithmically transformed (log CFU/g). Results are presented as mean values (±standard deviation) of the four (4) samples analyzed at each sampling point.

### 2.3. FT-IR Sensor

A ZnSe 45° HATR (horizontal attenuated total reflectance) crystal (PIKE Technologies, Madison, WI, USA) and an FTIR-6200 JASCO spectrometer (Jasco Corp., Tokyo, Japan) equipped with a standard sample chamber, a triglycine sulphate (TGS) detector and a Ge/KBr beam splitter was used for the acquisition of spectral data. A small quantity of each seaweed sample was cut into small pieces, placed onto the crystal surface and pressed with a gripper to ensure the best possible contact with the crystal. The crystal used shows a refractive index of 2.4 and a depth of penetration of 2.0 µm at 1000 cm^−1^. The spectra were obtained from the range of 4000 to 400 cm^−1^ by accumulating 100 scans with a resolution of 4 cm^−1^ and a total integration time of 2 min using Spectra Manager™ software version 2 (Jasco Corp., Tokyo, Japan). In total, 351 samples were spectroscopically analyzed, while 171 and 180 of them originated from MI and SAMS, respectively.

### 2.4. Multispectral Imaging—VideoMeterLab

The multispectral images were captured using the VideoMeterLab imaging system [17]. In essence, VideoMeterLab receives from each detector the surface reflectance of the samples at 18 different wavelengths ranging from 405 to 970 nm (405, 435, 450, 470, 505, 525, 570, 590, 630, 645, 660, 700, 850, 870, 890, 910, 940 and 970 nm), creating a data cube of spatial and spectral data for each sample. The sample was placed inside a sphere (Ulbricht sphere) with light emitting diodes (LEDs) with a narrowband spectral radiation distribution. During data acquisition, the diodes were strobing successively, resulting in a monochrome image with 32-bit floating point precision for each wavelength. VideoMeterLab accompanying software was used to extract the region of interest (ROI), pixelwise spectral information (i.e., spectrum of each sample pixel at the 18 image acquisition wavelengths) and other relevant statistics such as the spectral average intensity and the corresponding standard deviation. In this analysis, 167 samples from MI and 209 samples from SAMS (376 in total) were used.

### 2.5. E-Nose Sensor

A commercial electronic nose system (product of FOX 3000, Alpha MOS, Toulouse, France) equipped with 12 metal oxide gas sensors (MOS sensors) based on different sensing materials available in our laboratory was used. Gas sensors were located in two temperature-controlled chambers under a high temperature and zero humidity, and a purified air generator was used to provide carrier gas for cleaning sensors. The sensors array was comprised of 12 metal oxide semiconductor (MOS) sensors, while the main applications of those sensors are depicted in Table 1. Based on sensor coating materials, the LY sensors (LY2/G, LY2/AA, LY2/Gh, LY2/gCT1, LY2/gCT and LY2/LG) were p-type semiconductors, while the P and T sensors (T30/1, P10/1, P10/2, P40/1, T70/2 and PA2) were n-type semiconductors. Sensor response was recorded by Alpha Soft v.12 accompanying software (Alpha MOS, Toulouse, France). The sampling conditions (quantity, volume, temperature and headspace generating time) were optimized prior to data acquisition in order to improve the performance of the sensor.

Before data acquisition and downstream analysis, samples were cut into small pieces, while 1 g of each sample was placed in a 20 mL vial. Then, the samples were placed into a thermoblock at 50 °C for 20 min for headspace generation and 0.5 mL from the headspace was drawn off and injected into the electronic nose. Each sample was transferred to the detector at a constant rate over 120 s. Following that, a cleaning procedure of the detector chamber was carried out until the sensor signals returned to baseline. As sample gases flowed over the sensors, the sensors’ resistance (R) changed. Therefore, a ratio (R−R0)/R0 was used to estimate the changes in sensor resistance, where R0 is the sensor’s resistance baseline and R is the real-time resistance. Twelve maximum response values of each sample from each sensor (Table 1) were extracted and further analyzed. In total, 291 samples were analyzed with the e-nose sensor (177 from MI and 114 from SAMS, respectively)

### 2.6. FT-IR Data Analysis Pipeline

A detailed description of the data analysis workflow for the FT-IR sensor is provided herein. In brief, the processing pipeline consists of feature selection, i.e., specific wavenumbers, on the basis of random forests (RFs) regression ensemble [18], followed by partial least squares (PLS) regression coupled with automated selection of a number of components. First, the FT-IR spectra were truncated after their acquisition, from 4000–400 cm^−1^ originally, to 2700 to 900 cm^−1^. This way, the peak related to the water and the inherent noise, mainly apparent at the two extremes of the acquired spectra, is excluded. Prior to the feature selection process, the spectra are normalized under the robust normal variate (RNV) normalization scheme [19] (Equation (1)):(1)$$s_{i}^{snv}=\frac{s_{i}-\mathrm{median}\left(S\right)}{\mathrm{mad}\left(S\right)}$$
where *S* is the ensemble of all spectra, and *s_i_* and $s_{i}^{snv}$ are the *i*th and the corresponding normalized spectra, respectively. Median absolute deviation (mad) [20] is a robust measure of the variability of a univariate sample of quantitative data *s*_1_, *s*_2_,…, *s_n_* computed as: (2)$$\mathrm{mad}=\mathrm{median}\left(\left|s_{i}-\mathrm{median}\left(S\right)\right|\right)$$

RNV, apart from enhancing data quality, reducing the correlated information across the different wavelengths/wavenumbers and eliminating the multiplicative noise inherent in to the acquisition process, produces results that are artifact-free, leading to improved downstream analysis. Afterwards, a feature selection step was introduced. Feature selection, as a pre-processing data procedure, has been found to be critical for various regression/classification problems with high dimensionality, where a low number of samples with a large number of variables (i.e., high dimensional space) are available. In order to overcome this issue and not fall into overfitting issues of any machine learning approach considered, the variable/feature set has to be decreased in a way in which meaningful features are preserved, while irrelevant (to the prediction of microbial contamination) and redundant ones are excluded. As in Tsakanikas et al. [21], random forests [18] are employed. RFs are an ensemble learning method for a regression that constructs a multitude of decision trees at training time while providing an output of the mean prediction (regression) of the individual trees. RFs were trained using LSBoost (gradient boosting strategy applied for least squares) [22] and 100 learning cycles. The dataset has been split randomly into training and test sets using a random generator. The feature selection process resulted in a minimum number of features (~90 features from 1800 initially) in all cases. Next, PLS regression [23,24,25] was performed on the dataset of lower dimensionality. Relevant to PLS (and other machine learning approaches) and the significance of selecting the “optimum” number of variables, is the fact that the inclusion of data irrelevant to the phenomenon under study, herein the microbiological abundance, may lead to limited performance and efficiency of the resulting model. As in Tsakanikas et al. [21], the decision on the number of the PLS components was performed on the basis of the variance plot and the number of model coefficients known as the “knee-point” of the curve. Intuitively, this is defined as the intersection of two lines outlined during an iterative curve fitting procedure. The training process is performed on the training dataset with 10-fold cross validation and 10 Monte-Carlo repartitions of the data for model calibration. 

### 2.7. Μultispectral Imaging (MSI) and E-Nose Data Analysis Pipeline

In the case of the multispectral imaging (VideometerLab—MSI) and e-nose sensors, the features (data dimensionality) are limited compared to the FT-IR and to the number of the samples. More specifically, MSI had 36 features (18 mean values and 18 standard deviations of the values across all wavelengths) and for the e-nose, the features correspond to the 12 sensors’ values. Thus, it is apparent that feature selection was not as critical as it was in the FT-IR case. Nevertheless, a “soft” cleaning of the features in each sensor system was applied, excluding those that did not provide any information related to the phenomenon under study, i.e., microbiological abundance. To this end, we estimated the distribution of each variable/feature and investigated whether they exhibited a bimodal distribution along the samples. Any variable that did not have a bimodal distribution was excluded since no informative variation was present in terms of microbiological load for samples with low and high TVC values. This was applied by fitting to the data (values of each sensor across all samples) one mode and two mode Gaussian mixture models [26]. Then, and in terms of the calculated Bayesian information criterion (BIC) for these two cases, the GMM model with the lowest BIC value was selected as the one that optimally represented the underlying data. The variables/features that were selected for the downstream analysis were the ones that exhibited a bimodal distribution. Intuitively, the variables represented by one Gaussian distribution showed a population that exhibits only one phenotypic mode, while the two Gaussian mixture distributions showed a phenotype capable of having two extremes, low and high microbiological load, and intermediate modes as well. In the spoilage case we expected that the response of the sensors at each wavelength or individual variable in general would have a bimodal distribution (not clearly bimodal but broad enough to be considered the synthesis of two Gaussians) corresponding to the extreme case of very low and high contamination. It must be noted that the same pre-processing scheme as in the case of FT-IR was utilized for the case of MSI and e-nose sensors, i.e., RVN. Finally, the machine learning approaches applied are the ones described and presented earlier in the case of FT-IR.

## 3. Results and Discussion

### 3.1. Microbiological Results

The microbial counts from the Irish and Scottish seaweed cultivated in different harvest years and stored at different temperature conditions are presented in Table 2 and Table 3, respectively. As can be observed, the products presented a high variability in terms of microbial quality since the initial microbial load was significantly differentiated among the three harvest years in seaweeds of both origins. Apart from the different environmental conditions during the three years (different harvest period—late June in 2019, early June in 2020 and March in 2021), post-harvest procedures and treatments may have also contributed to these differences for both sites. Previous studies have also reported a high variability in the initial microbial populations of several seaweed species [27,28,29].

The storage temperature seems to play an extremely significant role in microbial growth and, subsequently, in the rate of products’ degradation, as was expected. This is evident in the products harvested in 2020, stored at four different isothermal conditions at 0, 5, 10 and 15 °C. For the products of both harvesting sites, microbial growth was remarkably delayed at low temperatures storage, whilst in seaweed from SAMS, the microbial population of samples stored at 0 °C were below 4.0 log CFU/g even after 10 days of storage. Sánchez-García et al. [30] reported high initial microbial cell counts reaching 8.5 × 10^7^ microbial cells mL^−1^. Throughout the storage period, this number increased significantly, being more pronounced in the samples stored at 15 °C compared to those stored at 4 °C.

The handling of seaweed immediately after harvest is of critical importance for the microbiological quality of fresh seaweed; it is important to keep the initial microbial load as low as possible in order to extend the products’ shelf life.

As far as microorganisms’ behavior is concerned, another point worth noting is the remarkable increase in the microbial population even within 24 h. By the time point microbial counts reached the level of 5.0–6.0 log CFU/g, a rapid increase in microbial load was observed, ranging from 2.0 to 4.0 log CFU/g in almost all of the tested products (in terms of origin, harvest year and temperature), rendering the products inappropriate for consumption. A summary of the microbiological load of MI and SAMS samples for all cases and storage conditions are presented in Table 2 and Table 3, respectively.

Considering all the above-mentioned, the determination of the shelf life should be prioritized in such products, taking into account factors such as species, environmental conditions and storage temperature.

Taking all the aforementioned into account, the high diversity of the samples analyzed with microbiological and sensor technologies, as showcased earlier, should be underlined. Therefore, in summary, the samples used for the analysis originated from two different aquaculture sites (Ireland and Scotland), were harvested in two (in SAMS case) and three (in MI case) different years and were of different microbial quality (during data acquisition) due to storage under different temperature conditions. This high variability can definitely add value to the performed analysis since realistic conditions were simulated; on the other hand, this makes the prediction of microbial populations a hard task, garnering support for the use of generalized and robust prediction models.

### 3.2. FT-IR Results

Typical deconvoluted FT-IR spectra of fresh and spoiled samples from both aquaculture sites (MI and SAMS) harvested in 2020 are presented in Figure 1. Deconvolution is a function that analyzes bands that may contain overlapping curves and distinguishes the peak position for each band. The region of the spectrum in the range of 1800 to 800 cm^−1^, can provide important information about changes occurring in specific functional groups, which are related to certain chemical groups (esters, aldehydes, alcohols, etc.) and molecules (proteins, lipids, carbohydrates). All of these groups, which are substantial components of the seaweed matrix, are subjected to changes after harvest and throughout storage which may affect the quality of the product (showing the degradation). Peak (1) at 1745 cm^−1^ (Figure 1) is related to the stretching C=O bonds of the esters of lipids and fatty acids, while the large broad peak at 1637 cm^−1^ is relevant to the water content (O-H stretching). It should be noted that peaks related to chlorophyl content (1660, 1653, 1638 cm^−1^) [31] are overlapping with this large water peak, making it impossible to take them into consideration for further analysis. Peak (3) at 1548 cm^−1^ is related to amide II and can be used for protein quantification. Each of the three wavenumbers, 1043 (12, MI), 1074 (11, MI), (10, SAMS) or 1156 cm^−1^ (10, MI), (9, SAMS) can be used for the quantification of carbohydrates [32]. Carbohydrates belonging to the polysaccharides group is partially of great importance in seaweed since there are certain functional compounds in this group such as alginates (1030, 1080 cm^−1^) and fucoidan, a sulphate polysaccharide frequently found in brown seaweed like *A. esculenta*. Bands around 1220 cm^−1^ (9-MI), (8-SAMS) are assigned to the presence of sulphate ester groups (S=O, which is a characteristic component in fucoidan and other sulphated polysaccharides. Additionally, peaks at 1415–1380 cm^−1^ (peak 5) are due to S=O stretching vibrations in sulphate. Brown seaweed is also known for its high content of phenolic compounds, which can be found near 1260 cm^−1^. Samples from both sites exhibited a peak at 1080 cm^−1^ (11-MI, 10-SAMS), which arises from C-O stretching of primary and secondary alcohols, while a similar peak at 1020 cm^−1^ (12-MI, 11-SAMS) is due to C-O stretching of primary alcohols [33]. Finally, peaks of weak bands in the range of 800 to 920 cm^−1^ (13, 14) can be assigned to C-H bending vibrations which can be found in monosaccharides, such as glucose and galactose [34].

The aforementioned functional groups, apart from being very important for the characterization of seaweed quality in general, are also of critical importance for estimating the microbiological quality as many of these compounds are expected to change during the spoilage phenomenon evolution. Certain molecules, such as sugars and proteins, are consumed by microorganisms and as microbes grow, several metabolites (alcohols, esters, aldehydes, sulphur compounds, etc.) are produced at the same time that can be imprinted on an IR spectrum.

The prediction results in terms of microbial abundance (i.e., TVC values) for the data acquired with FT-IR spectroscopy are presented below. The selected features/wavenumbers are presented in Figure 2, where it can be noticed that the critical information in terms of TVC estimation is not only located at wavenumbers where a “difference” is apparent between spoiled and fresh samples (e.g., area in [1000, 1100 cm^−1^]), since this area and the corresponding features are correlated not to the TVC values but to other sources of variation, e.g., origin, harvest time or other sources. The workflow followed in order to train and evaluate/validate the efficiency and performance of the prediction model has been described in the Section 2 (Methods Section). In order to obtain a strict efficiency evaluation, we moved as follows in terms of data separation into training and test datasets. The data for *A. esculenta* MI spanned over three years of collection, i.e., 2019, 2020 and 2021. For the first two years (2019 and 2020), the sample size was 107, from which ~25% of them, specifically 28 samples, were used as an external validation dataset and the rest, i.e., 79 samples, were used as a training set. The splitting was performed using a random generator to permit a fully randomized selection of the samples. Then, all data from the 2021 harvest was kept out of the training phase and along with the 28 test samples, forming a large test/validation set of 92 data samples. Table 4 (MI) and Figure 3 (A-MI, B-MI, C-MI) present the results of the linear regression between the actual, measured by microbiology techniques, and the predicted TVC values.

The developed model for the FT-IR sensor with MI origin samples exhibited good performance in terms of fitting results via the linear regression (i.e., y = αx + β) with slopes of 0.96, 0.92 and 0.83 for the cross-validation phase, test phase with the 28 samples from harvests 2019 and 2020 (A) and harvests 2019, 2020 and 2021 (B), correspondingly. The R-squared values (showing the goodness of fit) were 0.96, 0.90 and 0.63, while the RMSE values were 0.38, 0.86 and 1.50, respectively. All of the above indicate a good prediction performance of the model on the external test data sets and the “worsening” of the corresponding indicators is something that is expected due to the addition of a large data set consisting of a new batch of samples (from a different year). This dataset has not been “seen” by the model during the training phase, i.e., harvest samples from 2021. The observed decrease in performance was expected due to the additional variability of samples from a completely different harvest year and also the time period of harvesting within the year. Apart from the slight performance decrement, the model and, thus, the adopted methodology, is able to identify the features, the specific wavelengths of the spectrum that are the most information-rich and relevant to the phenomenon under investigation and the TVC values, i.e., microbiological spoilage of the corresponding product. Magwaza et al. [35] investigated the ability of PLS models using data from individual orchards and data combining orchards from two different harvesting seasons to predict physicochemical attributes of mandarin fruit. Fruits from different harvest seasons (2011 and 2012) were used to demonstrate the effect of seasonal variability on Please ensure your meaning is retained model prediction performance. The 2012 harvest season fruits were used in training models, while those from the 2011 harvest season were used in the testing procedure. The low prediction accuracy indicated that the seasonal variation significantly affected calibration models across seasons. In order to minimize the effect caused by seasonal and origin diversity, robust models combining two harvest seasons and four different orchards were developed. The robust model including more orchards and harvest seasons showed improved prediction accuracy compared to specific models developed based on a single orchard or single harvest season.

A similar approach for data splitting Into training and the test set was employed for *A. esculenta* samples from SAMS. In this case, the samples were not from a different harvest year so as to be used as the test set (like the 2021 harvest in *A. esculenta* from MI) due to the limitations of the scope for efficient training sample sizes. Samples from the two harvest years (2019 and 2020), accounting for 124 samples, were adequate for the model development. Using the same methodology as before, we trained a prediction model using 84 samples, while the rest of the 40 samples were used as the external validation data set. In Figure 3 (A-SAMS, B-SAMS) and Table 4 (SAMS), we summarize the results of the 1-1 fit of the predicted vs. the actual TVC values via a linear regression (i.e., y = αx + β), as shown previously. The results also indicated a good prediction performance of the model on the external test data set (data not used in training), although this resulted in a rather high RMSE. RMSEs up to 1.0 log CFU/g, although seemingly high for prediction bounds, are common in food microbiology even for state-of-the-art laboratory analysis, and thus, they are acceptable. It is well acknowledged that it is usually a deviation of the TVC value for the exact same sample when it is analyzed in parallel by two different persons.

Furthermore, for the above analysis and prediction model building, we moved forward towards building a model which would be independent of the seaweed origin and only look at the microbiological population estimation/prediction. To this end, a superset of data was created, consisting of both *A. esculenta* MI and SAMS. In this case, the data were split with a random generator following the “rule” of 75% (252 samples in total) as training and 25% (64 test data samples randomly selected) as test sets. This model, during the feature selection phase, is expected to disregard any features/wavenumbers closely related to the origin and their specific distinct way of spoilage due to the origin, leading to a model that would be origin agnostic with regard to seaweed spoilage estimation. The results for this approach with the superset of MI and SAMS *A. esculenta* data are shown in Table 4 (MI+SAMS) and Figure 3 (A-MI+SAMS, B-MI+SAMS). These results indicate a good prediction performance of the model in parallel with enhanced robustness for the two origins of seaweeds considered herein from the external test data set selected. It is apparent that the combination of samples from both aquaculture sites benefits the model development and its performance, making it origin-free for microbiological population prediction.

It is worth noticing that this work was conducted with samples stored at different temperature conditions. Apart from the differences originating from the metabolic products, different storage temperature can change the cellular structure of seaweed tissue and also affect absorptions in certain bands [36,37]. In general, spectra are sensitive to fluctuations in temperature conditions. It has been recorded that the weather conditions can change the sample temperature and subsequently change the spectra and affect the prediction accuracy [38]. The effect of temperature on IR spectra is related to the behavior of the hydrogen-bonded OH groups of water, which causes a broad absorption band around 1649 nm [39]. This broad band can be comprised of five component spectra of five different hydrogen bonds. The cluster size of hydrogen bonds decreases as the temperature increases and this can affect the relative absorbance value of the clusters with no hydrogen-bonded OH groups. Consequently, the hydroxyl band of water shifts to the lower wavelengths and becomes sharper as the temperature increases [40]. It is also indicated that this problem would arise in any other products containing a high water level (>80%), including seaweed. Therefore, the temperature fluctuations can affect both the spectral data acquisition and prediction models in real-world applications.

### 3.3. E-Nose Results

In e-nose instruments, electrochemical sensor arrays and proper identification equipment are employed. This differs from GC, GC-MS and other analytical methods in that it obtains not the qualitative and quantitative results of the individual components of the samples, but rather the overall information of the volatile components in the samples, that is, the fingerprint data [41].

Representative responses/signals of the e-nose sensors (highest intensity within the 120 secs of analysis) of fresh and spoiled samples of different origin stored at different temperatures are presented in Figure 4. “Fresh” samples are the average of 30 different samples with a population lower than 4.0 log CFU/g, while the ‘’spoiled’’ ones are the average of 30 different samples with a population higher than 8.0 log CFU/g.

In general, signals from fresh samples were stronger than those from spoiled ones for most of the tested cases. Additionally, samples from MI presented higher values compared to SAMS samples for the same microbial counts. Based on Table 1 and the applications of each sensor, the response of most of them was expected to change during storage. Additionally, it should be also noted that sensors with low intensities (LY sensors) may significantly contribute to the estimation of microbial populations as their signal seems to change from fresh to spoiled samples.

The prediction model development and validation of the e-nose sensor for the samples originating from MI and SAMS and harvesting during years 2019, 2020 and 2021 (MI only) is presented below. A dataset of 83 samples was used for the model development, while 30 samples (not used in model training) were randomly selected to form and serve as an external test data set. Figure 5 and Table 5 present the results of the linear regression (i.e., y = αx + β) between the actual and predicted by the developed prediction model values of TVC. These results did not indicate a very good prediction performance, at least not as good as in the case of FT-IR, as presented previously. A reason for this performance shortage could be that the variables/features measured do not provide a strong correlation to the phenomenon under study, i.e., microbiological spoilage. Furthermore, some additional data were accumulated, representing the 2021 harvest of *A. esculenta* MI. Since the model developed previously did not perform very well, we chose to rebuild the model using all harvests’ data samples and split them into training and test data sets from scratch, as described earlier. Thus, in total we accounted for 176 samples (years 2019, 2020 and 2021) where the external validation (test set) was set to 50 samples. In Figure 5 and Table 5, the linear regression results are shown (i.e., y = αx + β), showcasing a slope of ~0.76, an R-squared value of the fit of ~0.71 and an RMSE of the predictions of ~1.28. These numbers indicate a good prediction performance of the model on the external test data set. It is apparent that the data enrichment approach, with the additional data from harvest 2021, merited the model development. Thus, a robust model that is capable of predicting the microbiological population was able to be developed. In the case of *A. esculenta* from SAMS, since no data from the 2021 harvest year were available, 30 test samples were randomly separated from the pool of 113 total samples. Using this split of data, we trained and validated the prediction model and the results are presented in Figure 5 and Table 5, showing the results of the linear regression (i.e., y = αx + β) between the actual and predicted by the developed prediction model TVC values. These numbers did not indicate a very good prediction performance of the model, at least not as good as in the case of FT-IR or even the results from the e-nose sensors in the case of *A. esculenta* MI, as presented earlier. Further, as shown previously, the most probable issue may be the size of the data and their inherent variability. To support this claim, the augmented dataset with the 2021 harvest shown earlier yields a model that outperformed the model using just the data from the 2019 and 2020 harvest. Finally, using data from both MI and SAMS, a very poor prediction performance of the model was recorded, based on the performance indices. 

Many metal oxide-type sensors provide a primary response to certain chemical species along with a large number of secondary (weak) responses that overlap among various sensor types. These overlaps are probably differentiated among samples, increasing complexity, which might be confusing for the model training. However, a statistical learning method can be employed to teach the sensors to acquire the characteristics of each sample under investigation [42].

### 3.4. MSI Results

The profile of spectra acquired through MSI analysis and the differences among different years in the spectral profile of the samples are presented in Figure 6. The data—mainly bands in the visible spectrum—exhibited large variations among the three harvesting years, probably due to the period of harvesting or other specific conditions (environmental conditions or practices during and after the harvest) affecting the end-product, mainly at the visual range, i.e., the color of the samples, which is commonly independent of the microbiological load.

The prediction model development and validation for the MSI of *A. esculenta* from MI and SAMS samples, from the harvest years of 2019, 2020 and 2021 (MI) and the harvest years of 2019, 2020 (SAMS) are presented below. 

The developed model for the MSI sensor was applied to the 20 test samples of *A. esculenta* MI (out of an 84 total sample size). In Figure 7 and Table 6, the results for the linear regression (i.e., y = αx + β) are shown. A slope of ~0.49, an R-squared value of the fit of ~0.51 and an RMSE of the predictions of ~0.95 were estimated. Those numbers indicate a relatively good prediction performance of the model on the external test data set. In addition, model evaluation was also performed using some additional samples from the harvest of 2021. Thus, there were, in total, 167 samples (years 2019, 2020 and 2021), while the external validation (test set) consisted of 30 samples. The results are shown in Figure 6, where a slope of ~0.35, an R-squared value of the fit of ~0.22 and the RMSE of the predictions of ~1.01 were calculated. Again, the prediction performance of the model on the external test data set was not satisfactory. It has been found that the samples from different harvest years exhibit a large variation in their MSI spectra (please refer to Figure 6), suggesting that maybe the MSI is not suitable for efficient microbial population estimation, unlike the FT-IR sensor and, to some extent, the e-nose sensor, as shown previously, mainly due to their independence from the “color” of the samples that misleads the prediction model.

In Figure 7 and Table 6, the results of the linear regression (i.e., y = αx + β) are presented, between the actual and predicted by the developed prediction model, for the TVC values of *A. esculenta* SAMS. As shown in Figure 7 the results of the linear regression (i.e., y = αx + β) with a slope of ~0.56, an R-squared value of the fit of ~0.40 and the RMSE of the predictions of ~1.93 do not indicate a fair prediction performance (unlike during the training with cross validation), which can be justified by keeping in mind Figure 6 and the corresponding issue of the spectral differences among the years of harvesting.

In the case that data from SAMS and MI were combined, performance statistics values were slightly better than those models developed for each origin in separate (R-squared, 0.8; RMSE, 1.14). Probably by enlarging the size of data, the model was trained/learned better (good performance statistics in cross validation) and the differences in products among the differences in years were more successfully incorporated into the model, while the significance of the visual features (colour related) was degraded.

Another explanation for the low performance of the prediction models in the MSI sensor case, is the known issue of non-uniform distribution of light on the fruit and vegetable surface existing in almost all the imaging systems. Part of tissue with low intensity, especially the region near to the border, might even be wrongly considered as defective. Particularly for hyperspectral imaging, the bright spots in the central position caused by overexposing and progressive darkness at the edges enhance this phenomenon. This issue also results in a high spectral variability, which subsequently increases the complexity of the calibration model and decreases at the same time the universality, practicability and accuracy of the models.

The content of chlorophylls, anthocyanins and carotenoids and their proportion determine seaweed appearance and color and serve as attributes of quality since their content changes during ripening, maturation and storage. In addition, seaweed’s surface presents differentiations and pigments distributed non-uniformly on it. This results in variations of color and, consequently, in spectral reflectance and absorbance ability. Thus, for example, spectra originating from green color seaweed areas demonstrate the highest reflectance value due to their strong reflecting ability, while darker, brownish areas’ spectra hold a relative lower reflectance value [37,43]. It is apparent then that it is really difficult to distinguish samples with quality “defects” from samples with discoloring (no quality downgrade is apparent), thus influencing the defect recognition and segmentation [39], limiting in this way the performance and the ability of a model based on optical information to perform adequately.

Although this study attempted—to a large extent—to increase the variability of the dataset—taking into consideration that in order to increase the robustness of calibration models, it is necessary to have sufficient variability in the calibration set—poor model performances were still observed in some cases, where solid justification has been provided. Inserting additional stochasticity and variability into the calibration models improves their generalization and, consequently, their prediction efficiency and robustness concerning the application to new samples and in real-life applications. Usually, but not exclusively, it also led to an increase in the models’ complexity, which is one of the main reasons contributing to overfitting. Thus, others should be extra careful during model calibration so as to lower the problem dimensionality (feature selection) in such a way that features are not directly correlated with the problem at hand (herein known as the spoilage estimation), which would be eliminated as new samples are added. This way, previously “significant” features that were more correlated to other aspects (e.g., origin or harvest year) than the one under investigation (TVC herein), are excluded during the recalibration of the model. Therefore, as stated in Zhang et al. [44], viewed in terms of game theory, the balance between the specificity and universality of models should be paid more attention.

## Figures and Tables

**Figure 1 sensors-22-07018-f001:**
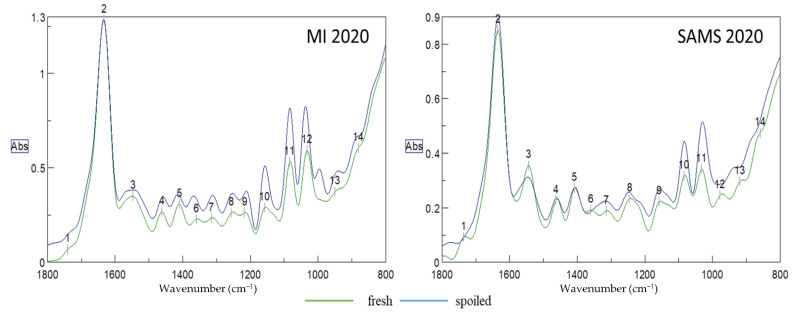
Representative deconvoluted FT-IR spectra of fresh and spoiled seaweed of different aquaculture sites harvested in 2020.

**Figure 2 sensors-22-07018-f002:**
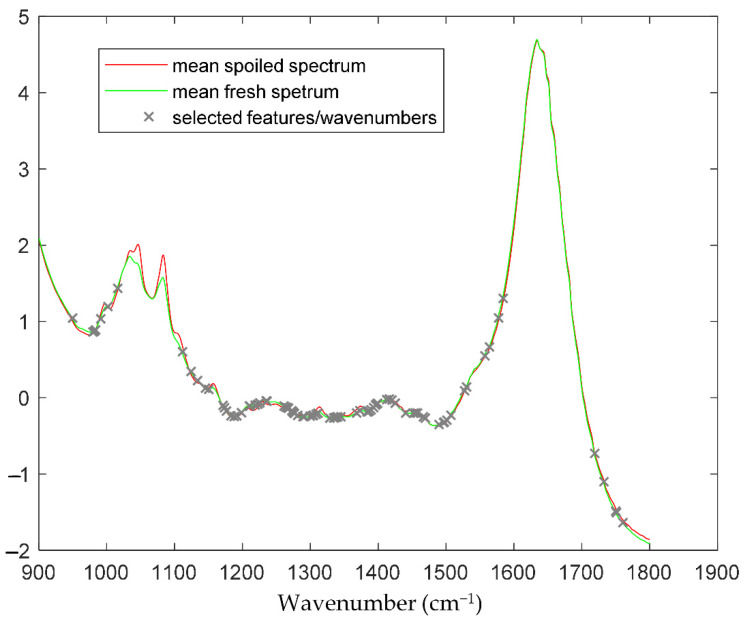
Average spectra for the fresh (green line) and spoiled (red line) of all samples (MI, SAMS and harvest year). Grey “×” symbols indicate the selected significant features/wavenumbers, ensuring that the wavenumbers implicated as the origin of the samples will not be taken into account by the developed prediction model.

**Figure 3 sensors-22-07018-f003:**
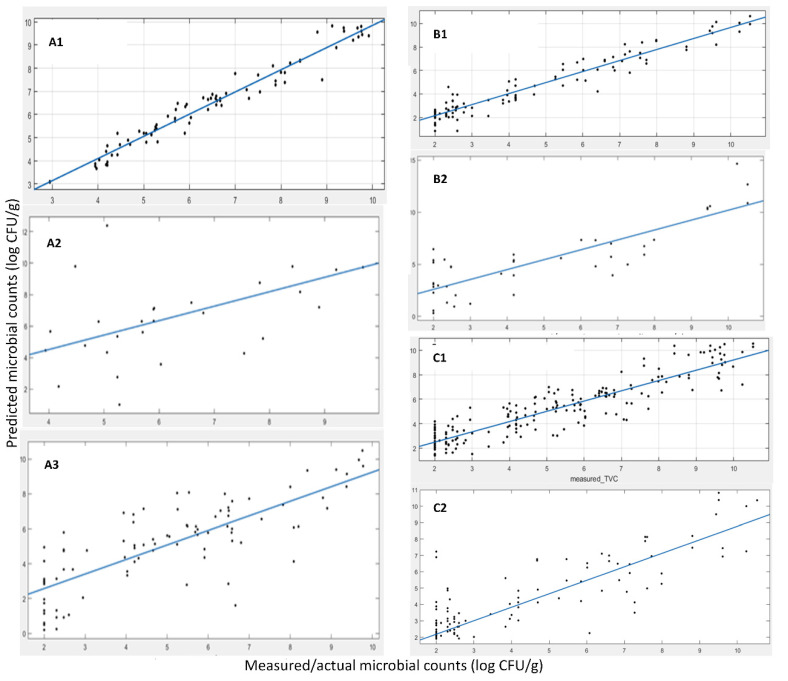
Linear regression between the predicted and the measured microbial counts using FT-IR data. (**A1**) cross validation MI samples, (**A2**) test/external validation from harvest 2019 and 2020 MI samples, (**A3**) test/external validation from harvest 2019, 2020 and 2021 MI samples, (**B1**) cross validation SAMS samples, (**B2**) test/external validation SAMS samples, (**C1**) cross validation, (**C2**) test/external validation MI+SAMS samples.

**Figure 4 sensors-22-07018-f004:**
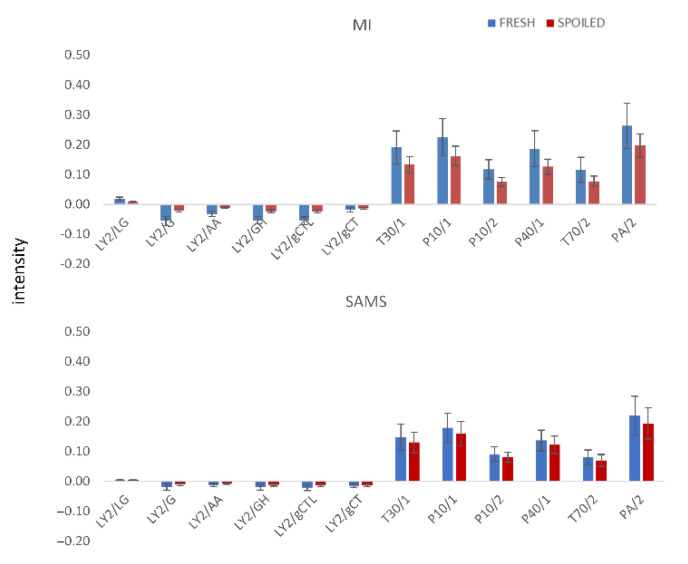
Maximum response values (resistance) from 12 MOS sensors of fresh and spoiled samples originating from MI and SAMS.

**Figure 5 sensors-22-07018-f005:**
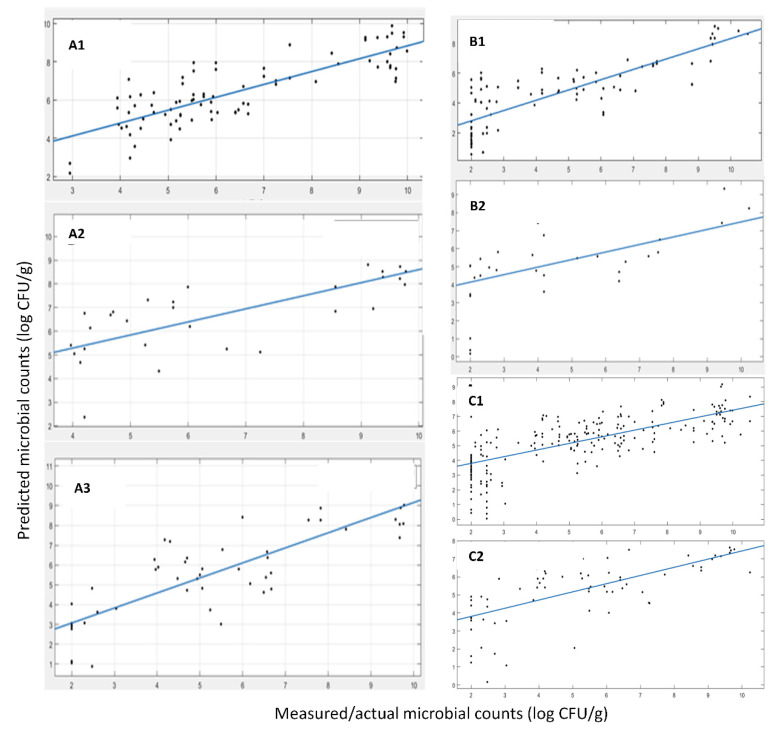
Linear regression between the predicted and the measured microbial counts using e-nose data. (**A1**) cross validation MI samples, (**A2**) test/external validation from harvest 2019 and 2020 MI samples, (**A3**) test/external validation from harvest 2019, 2020 and 2021 MI samples, (**B1**) cross validation SAMS samples, (**B2**) test/external validation SAMS samples, (**C1**) cross validation, (**C2**) test/external validation MI+SAMS samples.

**Figure 6 sensors-22-07018-f006:**
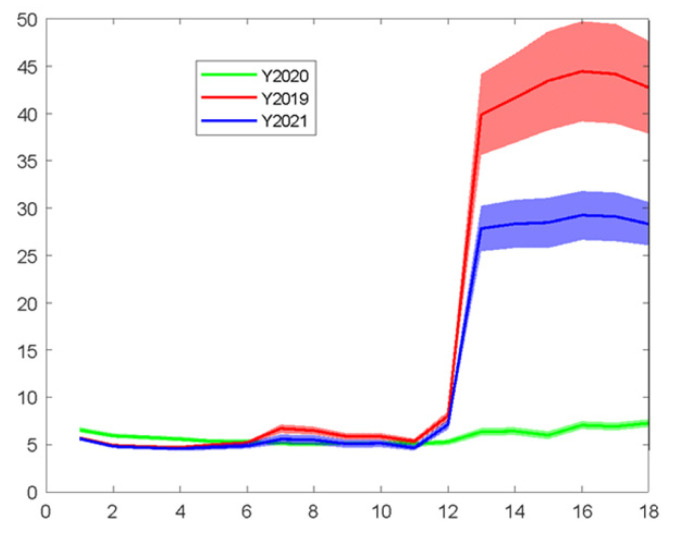
Differences of MSI spectra with respect to the year of harvest.

**Figure 7 sensors-22-07018-f007:**
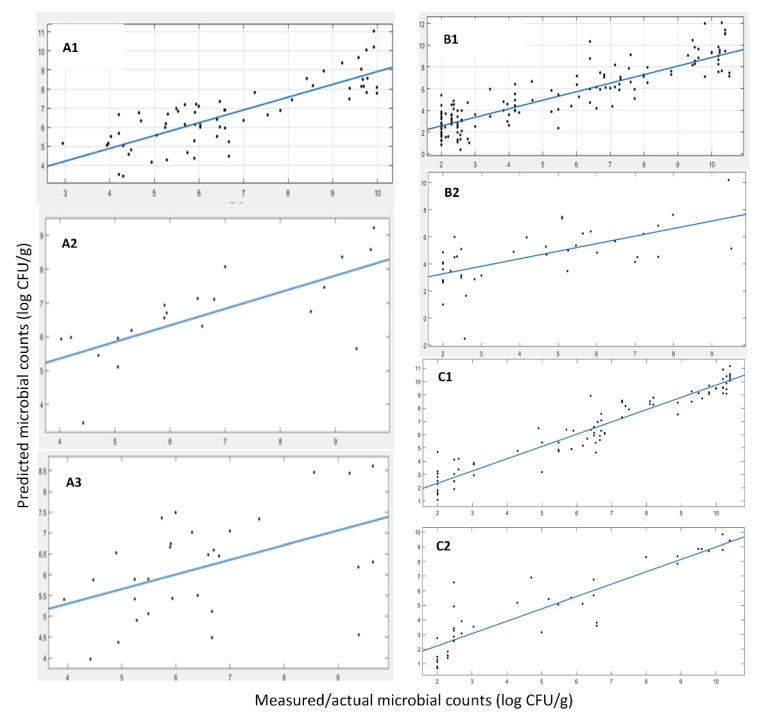
Linear regression between the predicted and the measured microbial counts using MSI data. (**A1**) cross validation MI samples, (**A2**) test/external validation from harvest 2019 and 2020 MI samples, (**A3**) test/external validation from harvest 2019, 2020 and 2021 MI samples, (**B1**) cross validation SAMS samples, (**B2**) test/external validation SAMS samples, (**C1**) cross validation, (**C2**) test/external validation MI+SAMS samples.

**Table 1 sensors-22-07018-t001:** Main application of 12 metal oxide semiconductor sensors in FOX3000, Alpha MOS e-nose system.

No.	Sensor	Application
1	LY2/LG	Oxidizing gas
2	LY2/G	Ammonia, carbon monoxide
3	LY2/AA	Ethanol
4	LY2/GH	Ammonia/Organic amine
5	LY2/gCTL	Hydrogen sulfide
6	LY2/gCT	Propane/Butane
7	T30/1	Organic solvents
8	P10/1	Hydrocarbons
9	P10/2	Methane
10	P40/1	Fluorine
11	T70/2	Aromatic compounds
12	PA/2	Ethanol, ammonia/organic amine

**Table 2 sensors-22-07018-t002:** Microbial population in seaweed harvested in 2019, 2020 and 2021 from the Marine Institute (Ireland) and stored under different temperature conditions.

		*Alaria esculenta*—Marine Institute											
Storage Temperature (°C)		0			5			10			15		
Harvest Year		2019	2020	2021	2019	2020	2021	2019	2020	2021	2019	2020	2021
**Storage Time (days)**	0	-	3.03 ± 1.87	-	4.90 ± 0.36	3.03 ± 1.87	2.30 ± 0.38	4.90 ± 0.36	3.03 ± 1.87	-	-	3.03 ± 1.87	-
	1	-	4.64 ± 0.90	-	5.20 ± 0.20	4.06 ± 0.47	2.46 ± 0.66	7.10 ± 0.41	6.16 ± 0.60	-	-	5.84 ± 0.22	-
	2	-	4.77 ± 0.41	-	6.00 ± 0.40	4.82 ± 0.67	2.30 ± 0.46	6.80 ± 0.59	8.12 ± 0.42	-	-	8.44 ± 0.50	-
	3	-	3.00 ± 1.36	-	6.70 ± 0.87	5.62 ± 0.45	2.30 ± 0.30	8.00 ± 0.48	9.66 ± 0.46	-	-	9.57 ± 0.27	-
	4	-	4.58 ± 0.82	-	7.10 ± 0.80	7.23 ± 0.93	3.80 ± 0.88	8.90 ± 0.14	9.67 ± 0.54	-	-	9.63 ± 0.47	-
	5	-	5.51 ± 0.43	-	7.50 ± 1.03	7.69 ± 0.98	4.50 ± 0.22	9.66 ± 0.24	9.65 ± 0.39	-	-	9.88 ± 0.15	-
	6	-	-	-	8.10 ± 0.20	-	-	9.98 ± 0.36	-	-	-	-	-
	7	-	6.46 ± 0.87	-	8.00 ± 0.54	7.61 ± 0.36	6.21 ± 0.59	9.54 ± 0.28	8.73 ± 0.29	-	-	9.74 ± 0.38	-

**Table 3 sensors-22-07018-t003:** Microbial populations throughout the storage of seaweed harvested from SAMS (Scotland) in two different years.

		*Alaria esculenta*—SAMS							
Storage Temperature (°C)		0		5		10		15	
Harvest Year		2019	2020	2019	2020	2019	2020	2019	2020
**Storage Time (days)**	0	-	1.80 ± 0.28	5.10 ± 0.86	1.80 ± 0.28	-	1.80 ± 0.28	5.10 ± 0.86	1.80 ± 0.28
	1	-	1.80 ± 0.28	6.40 ± 0.96	2.15 ± 0.41	-	1.65 ± 0.92	7.40 ± 0.46	2.18 ± 0.66
	2	-	1.91 ± 0.44	7.12 ± 0.34	1.00 ± 0.00	-	2.41 ± 0.58	8.56 ± 0.34	3.81 ± 0.52
	3	-	1.60 ± 0.22	8.20 ± 0.25	1.00 ± 0.00	-	2.63 ± 0.21	8.90 ± 0.56	5.27 ± 0.81
	4	-	2.48 ± 0.96	9.60 ± 0.20	3.27 ± 1.37	-	6.73 ± 0.47	9.80 ± 0.14	7.04 ± 0.32
	5	-	-	9.50 ± 0.46	-	-	-	10.24 ± 0.26	-
	6	-	3.39 ± 1.11	-	4.97 ± 0.41	-	6.30 ± 1.18	-	9.52 ± 0.12
	7	-	4.04 ± 1.88	10.30 ± 0.20	7.16 ± 0.80	-	8.20 ± 0.85	10.30 ± 0.20	10.01 ± 0.72
	8	-	-	-	8.62 ± 0.68	-	9.60 ± 0.65	-	-
	9	-	-	-	-	-	-	-	-
	10	-	3.98 ± 1.03	-	9.11 ± 1.59	-	-	-	-

**Table 4 sensors-22-07018-t004:** Linear regression fit parameters between actual and predicted TVC values for the different datasets (*A. esculenta* MI, SAMS, MI+SAMS) obtained from FT-IR analysis.

FT-IR				
**MI**	**α (Slope)**	**β (Offset)**	**R-Square**	**RMSE**
Cross validation	0.96	0.27	0.96	0.38
Validation A *	0.92	0.85	0.90	0.86
Validation B *	0.83	0.91	0.63	1.50
**SAMS**				
Cross validation	0.94	0.27	0.94	0.63
Validation	0.95	0.69	0.70	1.84
**MI+SAMS**				
Cross validation	0.84	0.82	0.84	0.96
Validation	0.82	0.53	0.75	1.77

* Validation A: 2019, 2020; Validation B: 2019, 2020, 2021.

**Table 5 sensors-22-07018-t005:** Linear regression fit parameters between actual and predicted TVC values for the different datasets (*A. esculenta* MI, SAMS, MI+SAMS) acquired from e-nose analysis.

E-Nose				
**MI**	**α (Slope)**	**β (Offset)**	**R-Square**	**RMSE**
Cross validation	0.67	2.11	0.67	0.97
Validation A	0.55	3.08	0.54	1.16
Validation B	0.76	1.55	0.71	1.28
**SAMS**				
Cross validation	0.69	1.42	0.62	1.30
Validation	0.42	3.31	0.50	1.46
**MI+SAMS**				
Cross validation	0.45	2.90	0.45	1.32
Validation	0.45	2.90	0.43	1.29

**Table 6 sensors-22-07018-t006:** Linear regression fit parameters between the actual and predicted TVC values for the different datasets (*A. esculenta* MI, SAMS, MI+SAMS) acquired from MSI analysis.

MSI				
**MI**	**α (Slope)**	**β (Offset)**	**R-Square**	**RMSE**
Cross validation	0.67	2.20	0.67	0.96
Validation A	0.49	3.40	0.51	0.95
Validation B	0.35	3.90	0.22	1.10
**SAMS**				
Cross validation	0.79	1.00	0.79	1.18
Validation	0.56	2.16	0.40	1.51
**MI+SAMS**				
Cross validation	0.92	0.50	0.92	0.81
Validation	0.84	0.54	0.81	1.14

## Data Availability

The data presented in this study are available upon request from the corresponding author.

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
