# Peer review of "Rapid Assessment of Microbial Quality in Edible Seaweeds Using Sensor Techniques Based on Spectroscopy, Imaging Analysis and Sensors Mimicking Human Senses"

_sensors, 2022, doi:10.3390/s22187018_

Round 1

Reviewer 1 Report

Seaweed production is a fast growing food sector and of great interest for health-food industry. Fresh seaweed products are highly perishable and a limited number of studies about determination of quality and safety are available. Therefore the authors investigated the potential of FT-IR spectroscopy, multispectral imaging and e-nose analysis combined with machine learning instruments for a fast prediction of microbial quality.

The sensor-based approach is very interesting and will contribute to facilitate the estimation of microbial counts, when the developed programme will ensure an acceptable outcome.

Remark to the design of the study: the different sampling sites and dates of sampling are comprehensible. But the differences of test runs regarding the harvest years and sampling sites were not explaned by the authors. different number of test runs will have an important impact on data viability and model performance.

minor remarks: the layout of table 2 and table 3 should be comparable. E.g. why different storage times? Table 2 seven days and table 3 ten days, also so number of columns in the table should be comparable for better comparability of the results.

Chapter 2.6.2 should be written in past tense

line 621: "...(no qulity...)" should be replaced by "... (no quality..)

Author Response

Response to Reviewer 1 Comments 

Seaweed production is a fast growing food sector and of great interest for health-food industry. Fresh seaweed products are highly perishable and a limited number of studies about determination of quality and safety are available. Therefore the authors investigated the potential of FT-IR spectroscopy, multispectral imaging and e-nose analysis combined with machine learning instruments for a fast prediction of microbial quality. The sensor-based approach is very interesting and will contribute to facilitate the estimation of microbial counts, when the developed programme will ensure an acceptable outcome. 

 Response: We really appreciate the effort and time spent by the reviewer(s) to review our submitted work and would like to thank them for their comments/suggestions towards elevating the quality of the manuscript. 

Point 1: Remark to the design of the study: the different sampling sites and dates of sampling are comprehensible. But the differences of test runs regarding the harvest years and sampling sites were not explaned by the authors. different number of test runs will have an important impact on data viability and model performance. 

Response 1: Each pilot site sent to Agricultural University of Athens one batch of seaweed per harvest year. Harvest time has been decided by the pilot site operators, taking into consideration parameters such as the size and the overall appearance of seaweed blades. Considering that the products have received frozen and they had to be thawed before analyses, we decided to do it once (4 technical replicates- line 119 in the revised manuscript) in order to avoid any changes on the microbial load due to freezing and thawing for a second time. These details have in added in the manuscript (please refer to lines: 93, 95-97, 106-108). 

Point 2: minor remarks: the layout of table 2 and table 3 should be comparable. E.g. why different storage times? Table 2 seven days and table 3 ten days, also so number of columns in the table should be comparable for better comparability of the results. 

Response 2: Thank you for your comment. With regard to your first raised comment, the different storage time periods were determined by the products’ microbial levels. At higher temperatures (10 and 15 degrees) microbial load was at high levels earlier during storage and consequently, there was no need for further extension of the storage time period. On the other hand, at 0°C (SAMS) the microbial populations were still at low levels even after 10 days of storage, while no further extension of the storage time was feasible due to the lack of extra samples. Concerning the number of the columns, this is due to the fact that for the Alaria esculenta - Marine Institute samples we had 3 years (2019, 2020, and 2021) of harvest while for the Alaria esculenta - SAMS 2 years (2019 and 2020). We preferred to keep the tables in their previous format so as to allow the readers of having a complete view of the experimental design and the samples used for the analyses. In 2019, 2 temperature conditions were selected since there was no information about the microbial behavior in these products during storage before (served as a screening phase). In 2020, products were stored at 4 different temperatures so as to have an overall view of the microbial growth in such products, while in 2021 (MI), the temperature of 5°C was selected as a typical refrigerator temperature condition.  

Point 3: Chapter 2.6.2 should be written in past tense 

 Response 3: We revised the corresponding section according to your suggestions. 

Point 4: line 621: "...(no qulity...)" should be replaced by "... (no quality..) 

 Response 3: Thank you, it has been corrected 

Reviewer 2 Report

Rapid assessment of microbial quality in edible seaweeds by using sensor techniques based on spectroscopy, imaging analy-3 sis and sensor mimic human senses

The work reports the application of spectral data from FT-IR and multispectral images and volatile compounds detected by e-nose to predict contamination in seaweed. The main novelty of the work resides in the comparison between the three techniques and the type of sample was it was applied. The work is well written overall, with some minor corrections being needed. The results are clearly presented and discussed, the material and methods are clearly described and are appropriate to the objective, and the data analysis is properly performed. I have only some minor suggestions regarding the background from the introduction and some information in the material section.

Introduction

The introduction is well written and organized. However, it only provides background for the quality and nutritional aspects of seaweed. I suggest the authors to provide some supportive information for the utilization of rapid analytical techniques (after lines 59-62). Since there are few works using these methods for seaweeds, as this is the major novelty of the work, I present here some suggestion of applications (only a suggestion, the authors may feel free to find other works if suitable) of these techniques in other non-conventional food products, to support the information. I believe that this would provide some further information for readers regarding the application of such methods:

Determination of protein content in single black fly soldier (Hermetia illucens L.) larvae by near infrared hyperspectral imaging (NIR-HSI) and chemometrics - https://doi.org/10.1016/j.foodcont.2022.109266

Applications of electronic nose (e-nose) and electronic tongue (e-tongue) in food quality-related properties determination: A review - https://doi.org/10.1016/j.aiia.2020.06.003

Detection and quantification of food colorant adulteration in saffron sample using chemometric analysis of FT-IR spectra - DOI: 10.1039/C5RA25983E

Chia (Salvia hispanica) seeds degradation studied by fuzzy-c mean (FCM) and hyperspectral imaging and chemometrics - fatty acids quantification - DOI: https://doi.org/10.17268/sci.agropecu.2022.015

Discrimination of edible oils and fats by combination of multivariate pattern recognition and FT-IR spectroscopy: A comparative study between different modeling methods - https://doi.org/10.1016/j.saa.2012.11.067

Also, it would be interesting if the authors provide one paragraph about the statistical methods used (PLS, etc).

Material and methods

Lines 82-87: please state clearly the number of samples analysed, as this is highly important to justify the multivariate statistical methods used. Also, please state how long samples were analysed after harvest, as it is not clear how long it took to transport from UK and Ireland to Greece.

Also, please inform whether the samples were dried and the approximate moisture content, as this may affect IR spectra. If not performed, please discuss the possible influence in the results and discussion section

I suggest the authors to inform whether the e-nose system was built by the research team, and the approximate cost, is this is greatly useful for further references that may apply this type of system, and is greatly innovative.

Lines 158-162: I suggest to remove this paragraph

Minor comments

The text should be revised for academic writing by an expert. Several sentences could be clarified. I hereby list a few of them:

Abstract

Line 19: please remove comma

Line 20: please remove comma

Line 23: please remove the first comma
